# A Feasibility Study of CT-Guided Osteosynthesis under Local Anesthesia

**DOI:** 10.3390/jpm13101493

**Published:** 2023-10-14

**Authors:** Joris Lavigne, Nicolas Stacoffe, Damien Heidelberg, Philippe Wagner, Jean-Baptiste Pialat

**Affiliations:** 1Department of Radiology, Centre Hospitalier Lyon-Sud, Hospices Civils de Lyon, 69495 Pierre-Bénite, France; nicolas.stacoffe@chu-lyon.fr (N.S.); damienheidelberg@gmail.com (D.H.); jean-baptiste.pialat@chu-lyon.fr (J.-B.P.); 2Centre D’étude des Maladies Osseuses, INSERM U1033, Université Lyon 1, 69008 Lyon, France; wagnerphilippe@yahoo.fr; 3Faculté de Médecine Lyon-Sud, Université Claude Bernard Lyon 1, 69495 Pierre-Bénite, France; 4Unité CNRS UMR 5220, INSERM U1294, Université Lyon 1, INSA Lyon, Université Jean Monnet, 42100 Saint-Etienne, France

**Keywords:** percutaneous osteosynthesis, interventional radiology, local anesthesia, pain

## Abstract

Background: Evaluation of local anesthesia for perioperative pain control in patients undergoing CT-guided percutaneous osteosynthesis. Methods: A total of 12 patients underwent percutaneous osteosynthesis under local anesthesia. Intraoperative pain was assessed after the procedure using numerical rating scale (NRS). Patients were also asked to rate their overall experience of the operation using the following scale: “highly comfortable, comfortable, hardly comfortable, uncomfortable” and, finally, “Would you be willing to repeat this intervention again under the same conditions if necessary?” Patients were also clinically followed up at 1 month, 3 months, and 6 months using the EQ5D5L scale to assess their pain and quality of life. Results: Patients underwent percutaneous osteosynthesis for osteoporotic (*n* = 9), pathological (*n* = 2), or traumatic fractures (*n* = 1), including the thoraco-lumbar spine (*n* = 8) or the pelvis (*n* = 4). The mean of NRS value experienced during the procedure was 3.4/10 (0–8). The overall feeling was highly comfortable (42%), comfortable (50%), hardly comfortable (8%), and uncomfortable (0%). Finally, 75% of patients answered “YES” to the question of repeating the operation under the same conditions if necessary. At follow-up, a significant reduction in pain was observed postoperatively. According to the EQ5D5L scale, there was a significant improvement in pain, mobility, self-activities, autonomy, and perceived quality of life at 3 and 6 months. Conclusion: Radiological percutaneous osteosynthesis is an effective technique that can be performed under local anesthesia with a comfortable experience for most of the patients.

## 1. Introduction

The combination of CT and fluoroscopic guidance in interventional radiology departments has enabled the development of minimally invasive techniques. Among these, there is a growing interest in interventional radiology for osteosynthesis using cannulated screws. There are several indications, such as traumatic fractures [1,2], insufficiency fractures [3,4], or pathological fractures [5,6], at the thoraco-lumbar, sacral spine, or at the pelvic bone mainly. The primary aim of stabilizing these lesions is to reduce the patients’ pain but also to reduce the time spent in bed and to enable the patient to return to sitting or standing position more quickly, further reducing the complications associated with prolonged lying. Some studies suggest that a three-dimensional guidance tool such as CT and a minimally invasive approach could reduce postoperative pain, hospital stay, and bleeding complications [7,8]. In addition, accurate screw positioning is an important factor in terms of biomechanical considerations to avoid post-procedural complications and material failure [9]. Thus, three-dimensional guidance would provide a high accuracy in terms of the screw positioning [10,11], which is superior to that provided by a two-dimensional guidance tool such as fluoroscopy alone [12]. Finally, the cost-effectiveness considerations support the use of CT-guided percutaneous osteosynthesis because it significantly reduces the need for a second procedure due to screw misplacement [13,14].

The percutaneous osteosynthesis technique (POS) using screws is now well established, with only a few variations from one institution to another. The materials used in the bony approaches are similar to those used in simpler procedures such as bone biopsy or percutaneous vertebroplasty [15]. Vertebroplasties performed under local anesthesia have been widely reported [16] and used in multiple centers for decades. Moreover, cases of spinal or pelvic osteosynthesis under local anesthesia have been reported by expert centers [17,18]. However, the feasibility of POS under local anesthesia has not been extensively studied. Nevertheless, some patients who are eligible for this type of procedure may not be able to benefit from general anesthesia or conscious sedation due to advanced age and numerous comorbidities. Unfortunately, these patients are the most prone to benefit from the osteosynthesis to quickly return to sitting or standing position. In order not to deprive these patients from this treatment, we have been led to propose percutaneous osteosynthesis under local anesthesia alone, following a case-by-case assessment of the benefit–risk balance. Our experiences and the preliminary data reported above led us to conclude that local anesthesia could be used to perform percutaneous screwing. In addition, since the COVID-19 period, anesthesiologists’ resources have been severely restricted; a number of patients were proposed to undergo POS procedure under local anesthesia. The objective of this study is to report the intraoperative pain and discomfort experienced by the patients and to assess the feasibility of POS procedures under local anesthesia.

## 2. Material and Methods

### 2.1. Population

Patients who underwent percutaneous osteosynthesis in our department between January and September 2020 were retrospectively included. The study was approved by the Ethic Committee of the Hospices Civils de Lyon (CNIL register number 20_5096). A letter was sent to patients informing them that the data needed for the study would be collected from their medical file. Patients with a recent fracture of the spine or pelvis of osteoporotic, traumatic, or pathological origin, who had undergone percutaneous treatment by osteosynthesis, with or not without cementoplasty, were included in the study. Patients were excluded if they did not consent to the use of their data collection for research purposes or if they were lost before the 6-month follow-up visit.

Patients were referred by orthopedic surgeons, oncologists, rheumatologists, or emergency physicians. The indication for percutaneous osteosynthesis was confirmed during a consultation or by a pluridisciplinary staff on the basis of the clinical and imaging findings. The fracture was confirmed by CT scan when the history revealed an evident recent trauma and by magnetic resonance imaging (MRI) when the recent aspect of the fracture needed confirmation.

Patients were informed of the indication, the course of the procedure, its objectives, and risks during a pre-interventional consultation by the interventional radiologist. Pain was assessed using a numerical rating scale (NRS) from 0 to 10.

### 2.2. Procedure

#### 2.2.1. Anesthetic Protocol

Premedication with 0.5 mg alprazolam and 20 mg immediate-release morphine was given 30 to 60 min prior to the procedure. Before setting up the patient in the operating room, 1 g of paracetamol was diluted in 20 mL of saline solution and injected intravenously.

#### 2.2.2. Screw Placement Procedure

Patients were placed in the prone position for the spinal and sacral bone screw placement or in the supine position for the pelvic bone approach. A slow intravenous injection of 2 g of cefazolin was given as antibiotic prophylaxis before the procedure. A CT volume acquisition was then performed to analyze the anatomy and to plan the entry point and the path used for the bony approach. After the patient was seated on the table, the skin was disinfected with an antiseptic solution based on 2% chlorhexidine in 70% isopropyl alcohol using sterile textile pads, followed by a one-minute drying period. The procedure was repeated a second time before the sterile surgical drapes were laid.

Superficial local anesthesia was achieved by injecting 3 to 5 mL of 1% lidocaine and 3 to 5 mL ropivacaïne using a 25-gauge needle. Anesthesia of deep soft tissues and periosteum was performed with 10 mL of the same mixture using a 20 cm, 19-gauge needle under fluoroscopic and CT guidance. Once the correct position of the needle was confirmed, the plastic top end of the 19-gauge needle was removed. The needle was then used as a wire to place an 11- or 13-gauge bone trocar. The size of the trocar was chosen according to the targeted anatomical location (6 to 15 cm). After removing the needle, the trocar was inserted into the bone under fluoroscopic guidance with manual pressure and rotational movements, especially when passing the fracture site. No mallet was used, in order to limit patient’s discomfort and to avoid the risk of fracture displacement. Once CT confirmed the correct trocar position, a 1.4 mm or 2.0 mm diameter Kirschner wire (K-wire) was positioned under fluoroscopic control (for 4.5 mm of and 6 mm diameter screws, respectively). The trocar was then removed, and the screw was mounted on the wire. Passing through the skin and muscle was facilitated by rapid successive clockwise/anti-clockwise 90° rotations of the screw on the wire. When the bone was reached, the screw progression was completed without applying excessive pressure to avoid bending the wire. Before removing the wire, the tip of the screw was anchored in the distal cortical bone to ensure mechanical stability.

In the case of a tumor lesion or an insufficient fracture in a poor-quality osteoporotic bone, an additional cementoplasty was performed using an additional 13-gauge trocar placed in the bone adjacent to the screw. Polymethyl methacrylate (PMMA) bone cement was injected under fluoroscopic guidance using a cement bone filler. When the bone was suitably filled, the trocar was removed. A CT scan was performed at the end of the procedure, to confirm the correct positioning of the screw and the correct bone filling by the cement. The patients were then monitored for 6 h in an inpatient room, with hourly assessments by a nurse of heart rate, blood pressure, pain on a numerical rating scale, and the appearance of the skin dressing to check for bleeding. The patient was allowed to stand and walk under the supervision of the radiologist at the first attempt, at least 2 h after the end of the procedure. Due to frailty, two patients were not able to stand initially, there were seated the next day. The follow-up consisted of a control CT scan at 1 month and a medical consultation at 1 month, 3 months, and 6 months.

### 2.3. Pain Evaluation

Immediately after the procedure, the patients were asked to rate the maximum intensity of pain during the procedure on a numerical rating scale (NRS) from 0 to 10. The patient then answered the two questions: “What was your overall feeling about the procedure, regardless of the pain?” on a 4-point scale: very comfortable, comfortable, hardly comfortable, uncomfortable and “Would you be willing to repeat this intervention again under the same conditions if necessary?”

Follow-up assessments were performed before the procedure and at 1, 3, and 6 months. Pain was assessed using the NRS and quality of life using the EQ5D5L scale (EuroQol Research Foundation, Rotterdam, The Netherlands) [19], an analogical scale evaluating on 5 levels by decreasing the following items: mobility, self-care, usual activities, pain/discomfort, and anxiety/depression, combined with a numerical score from 0 to 100% of the patient’s global health self-report.

### 2.4. Statistical Analysis

Patient characteristics and clinical data were collected from the patients’ medical records by the resident doctor, who was in charge of the patients. Data were entered into an Excel spreadsheet by the resident and then into an Access local database by a clinical research assistant prior to statistical analysis. Statistical analyses were carried out using the R software version 4.3.1 (Copyright © 2023 The R Foundation for Statistical Computing). The normality of the distribution for continuous variables was assessed using a quantile-quantile plot and then checked using a Shapiro–Wilk test. When the conditions were met, a paired *t* test was used. Otherwise, a non-parametric paired Wilcoxon test was used. Categorical variables were compared using a Chi-square test. When the validity conditions were not met, a Fisher exact test was used. The null hypothesis for the current research hypothesis was that POS would not be an effective technique that could be performed under local anesthesia, given the patient’s discomfort during the procedure and the non-significant improvement in pain and quality of life afterwards

Comparisons of the variables according to the different measurement times (baseline, 1, 3, and 6 months) were compared using a repeated-measures analysis of variance, and each follow-up was compared to the preoperative measure using paired *t*-test. Homogeneity of both variance and covariance were tested using the Levene’s test and the Box’s test, respectively. The tests were considered significant if the *p*-value was less than 0.05.

## 3. Results

Thirteen patients underwent POS procedures in our department between January and September 2020. Thus, no patient objected to the use of their data for research purposes. One patient was lost because she developed COVID-19 lung disease before the 3-month follow-up visit, which unfortunately led to her death. Eventually, 12 patients were included in the study, 6 males and 6 females (mean age 81 years, range 31 to 95 years), who received percutaneous screw fixation using one (*n* = 6) or two (*n* = 6) screws, combined with complementary cementoplasty (*n* = 11). The indications for POS were as follows: 9 insufficiency fractures; 8 vertebral fractures (4 lumbar and 4 thoracic) with osteonecrosis and monopedicular (*n* = 3) or bipedicular (*n* = 5) fracture. They benefited from vertebroplasty with either a single or a bilateral pedicle screwing. One sacrum fracture was treated by sacroplasty and transiliac screw osteosynthesis. One traumatic fracture of the acetabulum was treated with an anterior two-screw fixation and two pathological fractures, one fracture of the ilium–pubic branch on pulmonary metastasis, and one fracture of the acetabulum on myeloma was treated with cementoplasty and single-screw osteosynthesis (Figure 1). The characteristics of the population are shown in Table 1.

Before POS, background analgesic treatment was an opioid (morphine) in five patients, weak/low dose opioid in five patients, and paracetamol in two patients.

The mean NRS value of the pain rated during the procedure was 3.4/10 (range 0 to 8). The global experience was “highly comfortable” for five patients (42%), “comfortable” for six patients (50%), and “hardly comfortable” for one patient (8%). None of the patients reported that the procedure was uncomfortable. Eventually, nine patients (75%) answered “YES” to the question of whether they would perform the intervention again under the same conditions.

A significant reduction in pain was observed. The mean reduction after screwing in was 6 points at 1 month [95IC: 4.4–7.5], *p* < 0.0001., which was maintained at 3 months with a mean reduction of 5.5 points [95IC: 3.5–7.4], *p* < 0.0001, and at 6 months with a mean reduction of 7.2 points [95IC: 5.6–8.8], *p* < 0.0001, compared to preoperative values (Figure 2).

According to the EQ5D5L scale, there was a significant improvement in mobility, with a mean decrease in the disability score by 1.7 points at 3 months [95IC: 0.4–2.4], *p* = 0.02, and 2 points at 6 months [95IC: 0.9–3.1], *p* = 0.003. Similarly, the self-care score decreased by 1.2 points at 3 months [95IC: 0.2–2.2], *p* = 0.02, and 1.7 points after 6 months [95IC: 0.6–2.7], *p* = 0.01, and the disability in usual activities score decreased by 1.4 points at 3 months [95IC: 0.4–2.5], *p* = 0.01, and 1.4 points at 6 months [95IC: 0.3–2.6], *p* = 0.02. Anxiety/depression tended to decrease by 0.7 points at 3 months (*p* = 0.07) and by 0.8 points at 6 months (*p* = 0.08) (Figure 3).

Finally, the perceived global health was improved by an increase of 31% at 3 months [95IC: 5–56], *p* = 0.02, and by 37% at 6 months [95IC: 13–59], *p* = 0.006 (Figure 4).

Basic analgesic treatment was also significantly reduced in the postoperative with an average regression from a level 2 (weak opioid) to a level 1 (paracetamol) at 3 and 6 months (*p* = 0.01).

## 4. Discussion

The present study has shown that the percutaneous osteosynthesis under local anesthesia is well tolerated by most of the patients and leads to substantial improvement in pain and quality of life after the procedure. We decided to assess the feasibility of the POS by an NRS combined with a global experience’s assessment questionnaire. Indeed, it is important not to limit pain assessment to a numerical evaluation of the intensity felt during an interventional radiological procedure but to assess the patient’s global experience [16]. Intensity is one aspect of pain, but many other parameters such as pain duration, pain-related disability, and pain perception are also important to consider [20]. Furthermore, the acute intensity of pain is not sufficient to assess the impact of pain on a patient. Personal pain history, psychosocial factors, and the context of chronic pain, for example, also influence pain perception too, as many other studies have shown [16,21,22]. In our study, three subjects reported significant peroperative pain (7, 8, and 8/10 on NRS), but two out of three also reported a good overall experience. This supports the idea that acute pain intensity is only one component of pain. In addition, two patients had preoperative pain scores of 8 and 10/10 on the NRS, so they were patients with a significant overall pain condition that was not, or only slightly, exacerbated during the procedure. This also shows the tendency of these patients to rate their pain highly, yet dissociate it from the overall experience.

We did not precisely quantify the duration of the pain during the procedure, but most of the patients reported only brief pain peaks as the screw progressed, whereas they reported very little pain for most of the procedures. Some patients may find it difficult to tolerate the procedure due to the pain in the prone position. However, these patients were not offered such a procedure, as they were assessed during the preoperative consultation.

To the best of our knowledge, this is the first study to evaluate the tolerability of POS under local anesthesia; most of which have used conscious sedation in combination [17] or have been concerned with open surgical fixation of distal hand [23] or foot fractures [24]. In comparison with other interventional procedures under radiological guidance, two previous studies evaluated this tolerance in vertebroplasty [16,25]. In our study, the mean pain intensity evaluated by the NRS was 3.4/10, which was relatively close to those found in previous studies (5.5/10 or 5.7/10, respectively). Moreover, the proportion of patients who reported a very comfortable or comfortable experience in our study was 94%, compared with 75% [16] and 61% [25], respectively, which tends to indicate an overall good tolerability. In addition to the slightly more invasive character of POS compared with vertebroplasty, the difference between studies could be partly explained by the fact that the global experience rating [16] contained five scales, whereas our study included only four. The proportion of the response “comfortable” would probably have been lower with an additional scale. In addition, the two populations were not fully comparable due to an older population (81 vs. 70 years) and a higher proportion of tumor indications in our study.

Performing the POS procedure under local anesthesia has several advantages. First, the adverse effects of sedation or general anesthesia are avoided [26]. In addition, certain fragile patients are not eligible for general or conscious anesthesia, due to their age or significant comorbidity, which is common in oncological patients. Usually, they are excluded from surgical intervention [27], but they could still benefit from treatment by a POS under local anesthesia. Moreover, during the peak of COVID-19 pandemic period, access to anesthetic resources was very difficult. As POS was performed under local anesthesia, it allowed to reduce the organizational constraints and to save time, reducing the preoperative delay proposed to the patients and the duration of hospital stay. Another advantage is the real-time clinical assessment of the patient, especially during procedures with a high risk of neurological complications (e.g., on the spine or the sciatic nerve for instance). Thus, the procedure is safer. Eventually, the cost-effectiveness considerations support the use of local anesthesia compared with other techniques [28].

As POS is a procedure that can be proposed in a variety of indications, it is important to consider a number of factors before suggesting performing it under local anesthesia. These factors must be assessed during the preoperative consultation. The location of the fracture is an important parameter. A superficial location such as a vertebral pedicle [18] with a short subcutaneous and bony approach is a good indication because the operation will be shorter and easier to perform. In contrast, a deep site with a long bony trajectory, such as the sacroiliac joint, or a very oblique site, such as the ischiopubic ramus joint, will be more difficult to perform using local anesthesia without combined conscious sedation [17].

For the success of POS under local anesthesia, it is crucial also to assess the patient’s psychological state and stress level, and to determine whether the patient will be able to sustain the procedure or not. If the patient is receptive, talking to the patient helps to reduce stress and focus the patient’s attention on subjects other than the procedure being performed. In this way, the patient’s attention is diverted to pleasant things to hold on to for a better experience of the procedure. It is also necessary to assess whether the patient is able to maintain the position for an extended period without moving, especially in the prone position. Patients can have many causes of discomfort, whether related to the target site or another site. Associated lesions in the context of polytrauma may complicate the installation of the patient and positional tolerance. However, prolonged prone positioning may be difficult to maintain for degenerative neck pain, shoulder pain, or many other reasons. The good tolerability may also be partially related to aging. In fact, the periosteum, the bone layer with the most sensory innervation [29], becomes thinner and less innervated with age [29,30]. However, we lack data to study the influence of age. Our study included only one young patient (age: 31 years), who reported intraoperative pain as 3/10 on the NRS.

Systematic preoperative analgesic premedication helps to improve patient comfort during POS. In our center, we used alprazolam and immediate-release morphine 30 to 60 min before the procedure, according to the protocol defined with the anesthesiologists. Alprazolam is a benzodiazepine that acts as a central nervous system depressant by potentiating the activity of naturally produced GABA, used for its rapid anxiolytic effect. Morphine is an opioid analgesic that acts as an endorphin receptor agonist in the central nervous system. Their effects are complementary. In rare cases, gaseous analgesia using nitrogen protoxide [31] can also be used to supplement analgesic management, but none of the patients included in the study had this. Other methods, such as hypnosis techniques [32], the use of music [33], or virtual reality headsets [34], can help to protect the patient’s anxiety and improve comfort.

Eventually, patient peroperative comfort is also influenced by the type and size of the material used. Therefore, we believe that it is not reasonable to use a bone trocar larger than 11 gauge under local anesthesia, and it is even better to prefer 13 gauge if possible. The size of the screw is also important. For a 6.5 mm screw, the ideal wire size is 3.2 mm, but this requires an 8-gauge trocar. However, a 6.5 mm screw can be placed with a 2 mm wire, dropped through an 11- or 13-gauge trocar, but this will reduce the support of the wire. It is then preferable to initially limit the number of screws used in our POS under local anesthesia to one or two in order to keep the time within a tolerable range to limit the patient’s discomfort. However, reaching the learning curve, the procedure will be performed faster, allowing to position a greater number of screws in selected patients.

Our study has several limitations. First, it included a small sample size (*n* = 12), retrospectively, but, to our knowledge, this study is the only study investigating tolerability of POS under local anesthesia. The population is heterogeneous, including traumatic, osteoporotic, and pathological fractures, but the results are relatively suggestive. Secondly, the primary endpoint is subjective and retrospectively collected. In addition, the strong efficacy on pain and quality of life may lead to an overestimation of the good experience [16]. However, the most important factor is what remains of the intervention for the patient. Furthermore, we do not have a control group because it was impossible to ask a patient who underwent general anesthesia how they felt during the procedure.

Finally, there was a selection bias inherent in the patients and the conditions of the study, as many of the patients included were not eligible for general anesthesia, and it was very difficult to provide such anesthesia during the COVID-19 pandemic to those for whom it was not contraindicated. This was a bias but also a trigger for the study to show that we were able to offer the most appropriate therapy to the patients by controlling patient’s comfort during the procedures under simple local anesthesia, even in a very restrictive period for anesthesia.

## 5. Conclusions

Percutaneous osteosynthesis is an effective technique that can be performed under local anesthesia, with 94% of patients report being comfortable or highly comfortable overall experience of the procedure. This can be achieved by carefully selecting patients during the preoperative consultation, and by planning the procedure, using analgesic and anxiolytic pre-medication, taking care of the installation, using trocars no longer than 11 gauge, and using a maximum of one to two screws.

## Figures and Tables

**Figure 1 jpm-13-01493-f001:**
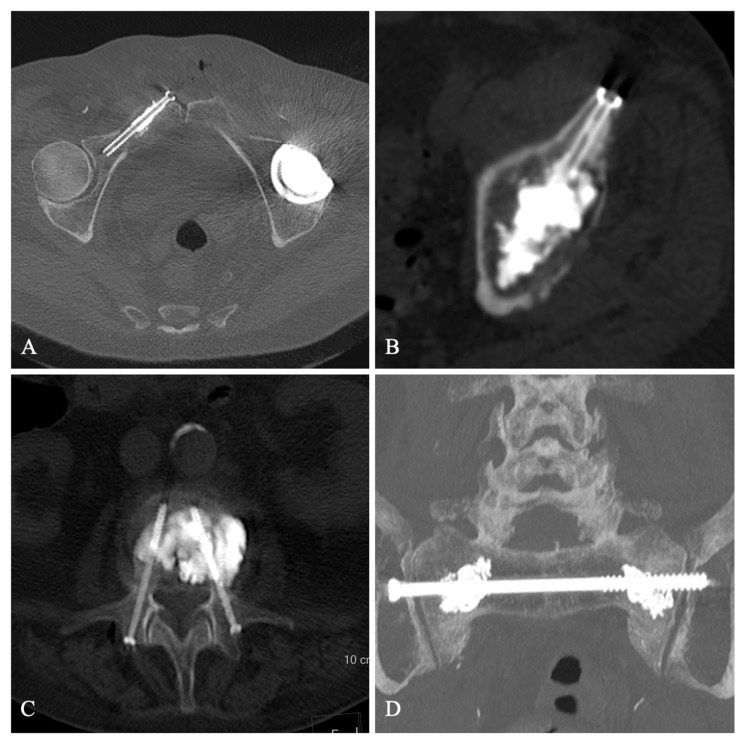
Screw fixation of a right iliopubic branch metastasis (**A**), Screw fixation of a myeloma lesion of the left acetabulum (**B**), Bipedicular screw fixation with cementoplasty of an osteoporotic vertebral fracture with osteonecrosis (**C**), Transiliac screwing with sacroplasty of an insufficiency sacral fracture (**D**).

**Figure 2 jpm-13-01493-f002:**
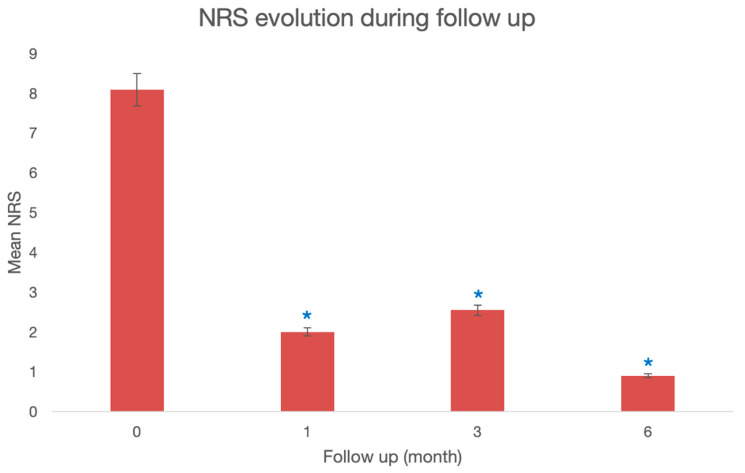
Median preoperative and postoperative numerical rating scale (NRS) between 1 day and 1 month, at 3 months and 6 months. *: Wilcoxon’s test compared to preoperative, *p* < 0.05.

**Figure 3 jpm-13-01493-f003:**
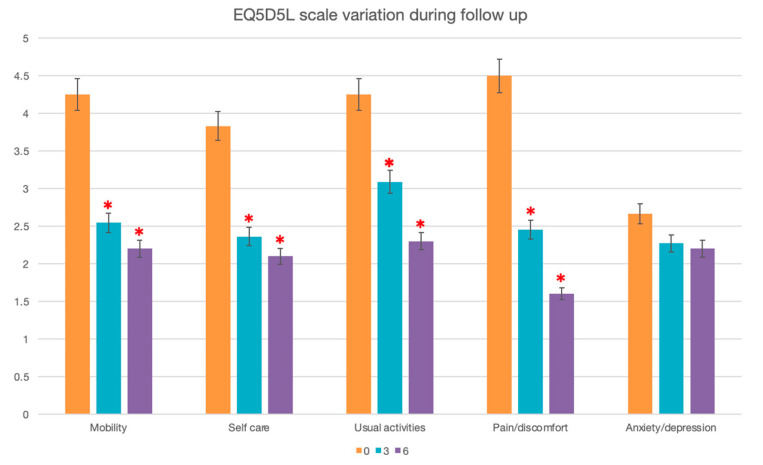
Median preoperative and post-operative at 3 and 6 months of mobility, self-care, usual activities, pain/discomfort, and anxiety/depression according to the EQ5D5L scale. *: Wilcoxon’s test compared to preoperative, *p* < 0.05.

**Figure 4 jpm-13-01493-f004:**
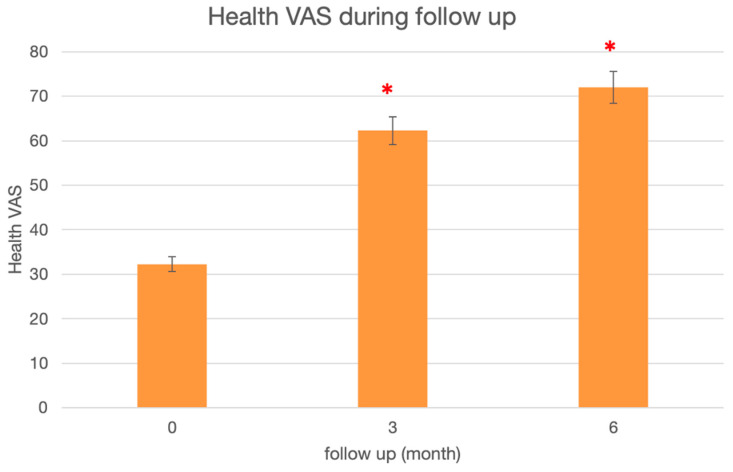
Median preoperative and post-operative at 3 and 6 months of global health by self-evaluation according to the EQ5D5L scale. *: Wilcoxon’s test compared to preoperative, *p* < 0.05.

**Table 1 jpm-13-01493-t001:** Population’s characteristics.

Demographic Characteristic	N (Number of Patients)
Age	
20–50	1
50–80	2
>80	9
POS * Indication	
Traumatic fracture	1
Insufficiency fracture	9
Oncologic fracture	2
Fracture localization	
Thoracic spine	4
Lumbar spine	4
Pelvic	4
Procedure variation	
Cementoplasty	
Yes	11
No	1
Screw	
1	6
2	6
Background analgesic treatment	
Paracetamol	2
Weak opioids	5
Opioids	5

* POS: Percutaneous Osteosynthesis.

## Data Availability

Data is unavailable due to privacy or ethical restrictions.

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
