# Peer review of "A Feasibility Study of CT-Guided Osteosynthesis under Local Anesthesia"

_jpm, 2023, doi:10.3390/jpm13101493_

Round 1
Reviewer 1 Report
Dear authors,
Thank you for submitting your work. Please address the comments raised and resubmit.
There are a number of small paragraphs (1-2 sentences) in the introduction and methods. Please organize your manuscript appropriately.
Mention how skin disinfection was performed.
The patient was then medically monitored- what parameters were exactly monitored and how frequently needed to be mentioned.
The correct terminology is numerical rating scale (NRS) and not numerical pain scale (NPS).
Statistical tests used should be elaborated- how the distribution of data was assessed, and how continuous/dichotomous data were analyzed.
Please mention how the data were entered (Excel, paper, any application/software).
If possible, present a flowchart of the included/excluded patients to summarize the results.
The English language needs to be addressed. There are a lot of technical and grammatical mistakes that need to be addressed prior to resubmission.
The English language needs major edits.
Reviewer 2 Report
Dear colleagues!
Your research is relevant and interesting.
In general, I have no serious comments, and please take the questions as a discussion that can improve the presentation quality of the work.
1. What is your null hypothesis?
2. Why did you use a numerical pain scale (NPS) rather than a visual analogue scale?
3. Anesthetic protocol: you use only one combination. Does this mean that the history taking involved selecting patients from the same anesthetic risk group, and also excluded the phenomenon of polypharmacy or drug cross-reaction?
4. Why was paracetamol chosen?
5. In the table you need to indicate in which units the calculation was made.
Round 2
Reviewer 1 Report
Dear authors,
The manuscript has been thoroughly revised and I can see the change in the write-up and the language in the revised version. I thank you for making changes as suggested.
The language has improved in the revised version.